# What do people living with chronic pain want from a pain forecast? A research prioritization study

Claire L. Little[1], Katie L. Druce[1], William G. Dixon[1,2], David M. Schultz[3,4]*, Thomas House[5], John McBeth[1,2]

1 Centre for Epidemiology Versus Arthritis, University of Manchester, Manchester, United Kingdom, 2 NIHR Manchester Musculoskeletal Biomedical Research Unit, Central Manchester University Hospitals NHS Foundation Trust, Manchester, United Kingdom, 3 Centre for Atmospheric Science, Department of Earth and Environmental Sciences, University of Manchester, Manchester, United Kingdom, 4 Centre for Crisis Studies and Mitigation, University of Manchester, Manchester, United Kingdom, 5 Department of Mathematics, University of Manchester, Manchester, United Kingdom

* david.schultz@manchester.ac.uk

**Data Availability Statement:** The datasets generated and analysed during the current study are not publicly available as data were collected for patient and public involvement activities and

## Abstract

Because people with chronic pain feel uncertain about their future pain, a pain-forecasting model could support individuals to manage their daily pain and improve their quality of life. We conducted two patient and public involvement activities to design the content of a pain-forecasting model by learning participants' priorities in the features provided by a pain forecast and understanding the perceived benefits that such forecasts would provide. The first was a focus group of 12 people living with chronic pain to inform the second activity, a survey of 148 people living with chronic pain. Respondents prioritized forecasting of pain flares (100, or 68%) and fluctuations in pain severity (94, or 64%), particularly the timing of the onset and the severity. Of those surveyed, 75% (or 111) would use a future pain forecast and 80% (or 118) perceived making plans (e.g., shopping, social) as a benefit. For people with chronic pain, the timing of the onset of pain flares, the severity of pain flares and fluctuations in pain severity were prioritized as being key features of a pain forecast, and making plans was prioritized as being a key benefit.

## Introduction

Chronic pain (i.e., pain lasting at least three months) is experienced by an estimated 43% of adults in the United Kingdom [1, 2]. Chronic pain conditions are associated with significant individual and societal burden. They are among the leading causes of disability globally [3], with an estimated 568 million global cases of lower-back pain alone in 2019 [4]. Individuals report that pain interferes with their professional and social lives, affects their relationships, and decreases their quality of life, mood and sleep [5]. In the UK, 13.4% of sickness days were due to musculoskeletal conditions in 2021 [6]. Although up-to-date figures are scarce, the economic costs of chronic pain are considerable. For example, chronic pain conditions cost 1.5–3% of European GDP in 2012 [7].

consent was obtained for the sharing of anonymous quotes and aggregated data only. The data within the Cloudy with a Chance of Pain study is detailed health data for a national population and is both sensitive and special category data. Given the detailed nature of the dataset, it is not possible to provide a minimal de-identified dataset that retains the necessary data utility to replicate our study's findings and be considered anonymised. Anonymisation of the Cloudy study data (whilst retaining data utility) is only possible through a combination of measures [i.e., de-identification, data minimisation related to the use case and the provision of access via a Secure Data Environment (SDE)]. These measures are in line with the UK Anonymisation Network guidance (Elliot, Mackey, & O'Hara, 2020). We are currently working towards establishing the processes for supporting access and sharing via an SDE and anticipate the data will be more widely sharable sometime in 2024. Elaine Mackey is the contact for dataset access (elaine.mackey@manchester.ac.uk).

**Funding:** This work was supported by infrastructure support from the Centre for Epidemiology Versus Arthritis (grant number 21755; https://www.versusarthritis.org/research/our-current-research/our-research-centres/ [versusarthritis.org]). TH receives funding from the Royal Society (grant number INF/R2/180067; https://royalsociety.org/ [royalsociety.org]) and the Alan Turing Institute for Data Science and Artificial Intelligence (https://www.turing.ac.uk/ [turing.ac.uk]). The funders had no role in study design, data collection and analysis, decision to publish, or preparation of the manuscript.

**Competing interests:** Coauthor Will Dixon has received consultancy fees from Google, and David Schultz has received consultancy fees from Palta, both unrelated to this work. All other authors have no conflicts of interest to declare. This does not alter our adherence to PLOS ONE policies on sharing data and materials.

The severity of chronic pain is a key driver of outcome, with more severe pain associated with worse outcomes including poorer physical and mental health–related quality of life [8–10], mood [11–13], and social and work participation [14, 15]. However, the absolute level of pain severity is not the only important driver of outcome. Variability in pain severity is also an important factor. The severity of chronic pain is not stable over time, and individuals experience intra- and inter-daily fluctuations in pain severity and pain flares which are characterized by a rapid increase in pain severity [16–22]. People living with chronic pain report that the variability in pain severity is unpredictable, and this unpredictability leads to feelings of uncertainty [23, 24] that permeates every sphere of their lives through a decreased ability to work, missed social events, and avoidance in making commitments [25, 26]. There is a clear desire to reduce the unpredictability of pain severity, with patients often asking how their pain might manifest in the future.

Pain is a complex biopsychosocial phenomenon and predicting variability in pain severity, including pain flares, will be challenging. It will involve identifying and understanding the complex relationship between time-varying biological, psychological and social exposures, discerning how those are associated with changes in pain severity over time, and developing models to forecast those changes. We propose that a personalized pain-forecasting model could reduce pain-related uncertainty by providing predictions of future pain. We have identified factors that are associated with variability in pain severity and could be used as predictors in such a model, including prior pain experience, physical activity [27, 28], mood [29, 30], sleep quality [31] and environmental exposures (here, the weather) [32].

Recent developments in digital data collection tools offer a solution to capturing these data. Patient-generated health data in chronic pain are already used to track daily symptoms including pain symptoms over time [33], to inform models of care [34] and to facilitate conversations between clinicians and patients [35]. Other spheres have shown the feasibility of using patient generated health data to forecast symptoms. For example, individualized prediction models exist for forecasting the diagnosis and prognosis of COVID-19 [36], the presence of anxiety and depression [37], the severity of hay-fever symptoms [38] and the level of physical fatigue [39]. It is feasible that patient-generated health data could also be used to forecast the variability in pain severity. However, the features that a pain forecasting model should predict are not yet clear.

There are many potential pain features that could be predicted by a pain-forecasting model including, for example, the level of forecasted pain severity described as an absolute value, the level of change in forecasted pain severity described as an absolute or proportional increase, the timing of that change, and the variability in pain severity over time. It is not clear which, if any, of these features people living with chronic pain would prioritize in a pain forecast. Patient and Public Involvement (PPI) is defined as work done *with* members of the public and can be conducted to involve stakeholders in the research process, including in identifying research priorities [40, 41]. Conducting PPI in the process of developing a pain forecast would ensure that the forecast is suited to the needs and priorities of its users [42]. Thus, identifying and prioritizing pain features in PPI activities forms the first stage in producing a pain-forecasting model.

The objectives of this work were to design the content of a pain forecast by (1) learning participants' priorities in the features of pain severity provided by a pain forecast and (2) understanding the benefits that participants perceive they would gain from such a forecast.

## Materials and methods

Two PPI activities were conducted with individuals with chronic pain. The first PPI activity was a focus group to inform the second PPI activity, a survey of people living with chronic

pain. The aim of the focus group was to identify potential pain features that could be produced by a pain forecast and a list of potential benefits of a pain forecast. The aim of the survey was to prioritize these features and benefits in a larger sample of people living with chronic pain. These PPI activities were approved by the Proportionate University Research Ethics Committee at the University of Manchester (Ref: 2021-11862-19751). The activities are reported in line with the GRIPP2 (Guidance for Reporting Involvement of Patients and the Public) checklist [41].

## Focus group

A semi-structured focus group was conducted with individuals with chronic pain to produce a list of meaningful pain features that could be provided by a forecast and to understand the perceived potential benefits of a forecast. A focus group was chosen as it allowed us to explore reasons behind the choices and to allow participants to build on each other's ideas [43]. A single focus group was conducted due to time and budget constraints.

We sought to recruit up to 12 individuals who were at least 18 years old, who self-reported having a noncancer chronic-pain condition, lived in the UK, and could read English. Participants were recruited through social media and shared through professional social media accounts of colleagues. We also asked charity organizations related to noncancer chronic-pain conditions (Table 1) to share the survey information through their newsletters and social-media channels, although we do not know if they did, how often they did, and how many people received that information. Potential participants completed a screening questionnaire, providing demographic information on their gender, ethnic group, age bracket (18–25, 26–45, 46–65 or 66+), self-reported chronic-pain condition(s) from a multiple-choice list and length of time since diagnosis. Participants for the focus group were then selected using purposive sampling, ensuring variation in age, gender, ethnic group, number and type of chronic-pain condition(s) and time since diagnosis. Recruited individuals provided informed written consent and were reimbursed for their time and expenses, in line with PPI guidelines from the National Institute for Health and Care Research [44].

The focus group took place in August 2021 and lasted approximately 90 minutes. Due to the ongoing COVID-19 pandemic, the focus group was held online using Zoom. Three

**Table 1. Charity organizations that advertised the study.**

| Charity organisation | Method of advertisement |
|---|---|
| Action for ME | Website, Twitter, Facebook |
| Arthritis Action | Website |
| Arthritis and Musculoskeletal Alliance (ARMA) | Newsletter, Twitter |
| Backcare | Website |
| Lupus UK | Online forum |
| Migraine Trust | Website, Twitter, Facebook |
| National Axial Spondyloarthritis Society (NASS) | Website, Newsletter |
| National Rheumatoid Arthritis Society (NRAS) | Website, Twitter, Facebook, Instagram |
| People in Research | Website |
| Postural Tachycardia Syndrome UK (PoTS UK) | Twitter, Facebook |
| PsAZZ Support Group | Contacted network |
| Scleroderma & Raynaud's UK (SRUK) | Twitter, Facebook |
| The Erythromelgalgia Warriors | Website, Twitter, Facebook |
| UK Gout Society | Twitter |
| Vocal | Contacted network |

**Table 2. Planned structure of the focus group.**

| Section of focus group | Purpose | Example questions/statements | Anticipated duration (mins) |
|---|---|---|---|
| Introduction and general overview | • Greeting<br>• Use of Zoom<br>• Review of ground rules<br>• Ice breaker | • Zoom: how to use hands up function<br>• We sent around some ground rules as suggested by the university. Is there anything that is missing that you would like to add? | 20 |
| Introduction to our ideas | • Outline the context of research<br>• Explain proposed research | Presentation | 5 |
| Breakout rooms | • Understand initial thoughts about the research and perceived interest in different predictands | • Here are some common patterns of pain severity. Which one(s) do you relate to?<br>• What pain features would you want to know about? | 15 |
| Break | | | 10 |
| Group discussion | • Bring thoughts from breakout rooms together | • What pain features did you come up with?<br>• Were there any in other groups that you think are good that you hadn't thought of? | 15 |
| Building a questionnaire | • Outline of a questionnaire that would be meaningful | • We want to build a questionnaire to ask other people what information they would like to know about pain patterns in the future. Here are some example questions that we have thought of<br>• Are there any questions here that you think we should remove?<br>• Are there any questions that we should include?<br>• What possible answers should we have?<br>• What information would you add or remove to the images to make them clearer? | 20 |
| Conclusion | • Thank participants & let them know what the next steps are | | 5 |

researchers (authors CL, KD and JM) co-facilitated the focus group and made written, anonymized field notes.

The structure of the focus group is provided in Table 2. Discussion topics focused on the pain features of a forecast that participants identified as potentially beneficial, how a forecast could be used in day-to-day life and how the survey (the second PPI activity) should be structured. These discussions led to revisions of the survey, details of which are provided in the results section. The structure of the focus group included a general group-level introduction (facilitated by CL), breakout-room discussions (facilitated by CL, KD and JM), and a final group-level discussion.

Group-level discussions in the focus group were audio-recorded through Zoom, without video recordings, and subsequently transcribed verbatim by CL. Field notes from breakout rooms were made by all facilitators. Participants' views on potential features of a pain forecast, and how a forecast may be used in day-to-day life were noted and subsequently used to inform multiple-choice questions in the survey.

## Survey

The second PPI activity was a survey of people living with chronic pain. This survey aimed to learn participants' priorities regarding the potential features and perceived benefits of a pain forecast, using the features and benefits identified by the focus group participants.

The survey was distributed in October and November 2021. The recruitment strategy was identical to that of the focus group. Members of the focus group were not prohibited from completing the survey. Consent was provided electronically at the start of the survey,

and only complete surveys were analyzed. We aimed to receive at least 100 completed surveys.

The survey collected demographic information and included priority-setting questions and multiple-choice questions. The demographic information collected were participants' gender, age, chronic pain condition(s) and site(s) of pain. Priority-setting questions asked participants to order multiple-choice options in preference order, using a drag-and-drop feature. Multiple-choice questions asked participants to select one or multiple options, with free-text space for further elaboration if required. Priority-setting and multiple-choice questions were written following discussions in the focus group, and so details are deferred to the Results section. No formal validation on the survey questions were performed. More generally, however, these questions and the answer selections on the survey arose from discussions within the focus group. That should have helped ensure that the questions were informed by conversations with people living with chronic pain. The survey was created in Qualtrics, and a link was shared on social media and with the charities listed in Table 1. The full survey can be viewed in the S1 Data.

Analysis of the survey questions was conducted descriptively. For priority-setting questions, the distribution of respondents prioritizing each option as most important to least important was calculated. For multiple-choice questions, the number and percentage of participants selecting each response was calculated. Sensitivity analyses using chi-squared tests were conducted to compare the responses of participants with commonly reported pain conditions to those without. No matching to the sample characteristics was performed during the analysis. Due to the small number of free-text responses, they were not directly analyzed but are reported following the relevant questions.

## Results

### Focus group

Demographic data of the 12 participants are provided in Table 3. There were nine females and three males, with all age brackets (18–25, 26–45, 46–65 or 66+) represented. Six participants had been living with chronic pain for at least five years, and five participants had two or more chronic-pain conditions. Chronic-pain conditions of the participants included osteoarthritis, chronic headache, fibromyalgia, neuropathic pain, rheumatoid arthritis, and spondyloarthritis.

Discussions in the focus group were centered on three overarching themes: the pain features, if any, that participants wanted a forecast to provide, the perceived benefits of a forecast, and the potential drawbacks of using a forecast in daily life. These themes are discussed in more detail below, followed by the implications they had on the survey questions.

The first overarching theme that participants discussed was the pain features of a forecast. Participants described pain features of interest in relation to periods of pain flares (when pain severity increased for a number of days) and periods of low or no pain severity (when pain severity decreased for a number of days). First, they spoke about the timing of these periods (such as when a period of low pain might begin), as this would have implications on activities that could be undertaken, and may impact how work and social activities could be managed in advance:

> P6 (F66+): "I gave an example of a [colleague] at the moment who's got a frozen shoulder and she'll be coming back on a phased return but the timing of that may be helpful through the [forecast]"

**Table 3. Demographics of focus group participants (*n* = 12).**

|  | Number of participants |
|---|---|
| **Age** |  |
| 18–25 | 1 |
| 26–45 | 3 |
| 46–65 | 6 |
| 66+ | 2 |
| **Gender** |  |
| Female | 9 |
| Male | 3 |
| **Ethnicity** |  |
| Asian British | 1 |
| Black British | 1 |
| British Chinese | 1 |
| South Asian | 1 |
| White British | 7 |
| White British & Black African | 1 |
| **Number of chronic pain conditions** |  |
| 1 | 7 |
| 2 or more | 5 |
| **Chronic pain condition*** |  |
| Chronic headache | 2 |
| Fibromyalgia | 2 |
| Neuropathic pain | 2 |
| Osteoarthritis | 5 |
| Rheumatoid Arthritis | 2 |
| Spondyloarthritis | 2 |
| Other | 3 |
| **Time since diagnosis** |  |
| Under 5 years | 6 |
| 5 or more years | 6 |

*Column exceeds 100% since participants can have multiple chronic pain conditions

Second, they discussed the duration of pain severity at high or low levels. Knowledge about the duration of pain severity at certain levels would also allow participants to plan activities and relevant interventions. For example, a participant in a breakout room commented that they may visit a chiropractor if their pain was predicted to last for a substantial period of time.

Third, participants wanted to know information about their pain-related quality of life and other symptoms (such as fatigue). Participants in breakout rooms noted that these other symptoms were not always correlated to pain severity and wanted to know information about them separately.

> P9 (F46–65): "I think you should maybe focus on quality of life. We all have different levels of pain, different levels of fatigue, we are all different. But what is important for me is totally different to the next person... It's on what you would accept as a quality of life"

The second overarching theme that participants explored were the perceived benefits of a pain forecast. The most commonly reported benefit of a forecast was that it could improve

the ability to make plans, including meeting family, planning grocery shopping, planning pharmacological interventions, organizing nonpharmacological interventions and adapting work:

> P3 (F26–45): "For me. . . the biggest advantage would be planning medication. . . I tend to go for the lowest meds, and then regret it because I'm still in pain and, oh God, now I can't take this, or I could take it but then I have stomach issues, all the rest of it. So. . . if I could get my drugs more accurate to how it's going to be, my pain medication, my PMR [steroid medication], that would really help, I think."

Another reported benefit was that participants hoped a forecast might support them in understanding triggers of pain, including how variables such as weather, stress and exercise might affect pain severity. In breakout rooms, participants discussed the empowerment granted through understanding their own triggers of pain.

In the third overarching theme, participants identified potential drawbacks of a pain forecast. First, participants highlighted potential mental-health challenges, such as anxiety and stress induced by having information about pain events, including pain flares:

> P3 (F26–45): "There are mental health disadvantages like anticipatory anxiety if the [forecast] tells you you're going to feel rubbish in a week."

Among these concerns, participants voiced fears of a self-fulfilling prophecy if they expected pain severity to increase.

Other participants highlighted mental-health challenges related to inputting pain-severity scores, which may encourage higher focus on the pain that they are trying to manage.

Second, participants were anxious about the potential implications of data collection during a pain forecast and fears of data sharing with employers and government officials:

> P7 (F46–65): "Who has access to the data? I think it would put a lot of people off if people thought that employers are going to have access to this data."

> P6 (F66+): Would it be "used by occupational health departments in organizations?"

Based on the discussion of the focus group, priority-setting and multiple-choice questions for the survey were written. Of the three overarching themes, questions were developed regarding the potential features and benefits of a pain forecast. As the drawbacks related to the implementation of a pain forecast, these were not included in the present survey. All suggested pain features of the focus group discussion were included, asking survey participants to prioritize the importance of timing, duration and symptoms during periods of low pain and pain flares. All reported benefits were included, asking survey participants to select which benefits applied to them regarding planning, applying pharmacological and nonpharmacological interventions and understanding triggers of pain.

## Survey

There were 148 respondents to the survey. Demographic information and data regarding chronic pain condition can be found in Table 4. Of the 148 respondents, 134 (90.5%) of respondents were female and 101 (80%) were aged between 36 and 65. The most commonly reported chronic-pain conditions were fibromyalgia (68, or 46%) and osteoarthritis (49, or 33%). Nine participants (6%) reported only 'other' pain conditions. These participants

**Table 4. Demographics and chronic pain condition of survey respondents (*n* = 148).**

| | | Number of participants | Percentage of participants |
|---|---|---|---|
| Age | 18–25 | 11 | 7.4% |
| | 26–35 | 19 | 12.8% |
| | 36–45 | 34 | 23.0% |
| | 46–55 | 27 | 18.2% |
| | 56–65 | 40 | 27.0% |
| | 66+ | 17 | 11.5% |
| Gender | Male | 13 | 8.8% |
| | Female | 134 | 90.5% |
| | Nonbinary/third gender | 1 | 0.7% |
| Chronic pain condition* | Fibromyalgia | 68 | 46.0% |
| | Osteoarthritis | 49 | 33.1% |
| | Chronic headache | 32 | 21.6% |
| | Neuropathic pain | 28 | 18.9% |
| | Rheumatoid Arthritis | 19 | 12.8% |
| | Spondyloarthritis | 17 | 11.5% |
| | Unspecific Arthritis | 12 | 8.1% |
| | Other | 69 | 46.6% |

*Column exceeds 100% since participants can have multiple chronic pain conditions.

reported ankylosing spondylitis, psoriatic arthritis, scleroderma, systemic lupus erythematosus, and juvenile arthritis.

Results of the multiple-choice question *"Which of the following would you like a pain forecast to provide for you?"* are shown in Table 5. The most commonly selected features were pain flares (100, or 68%) and fluctuations in pain severity (94, or 64%). Features of pain severity on an ordinal scale (70, or 47%) and periods of low pain (51, or 35%) were less commonly selected.

Recognizing that a large proportion of our respondents reported fibromyalgia (46%) and osteoarthritis (33%) as a chronic pain condition, we conducted sensitivity analyses to compare (1) the responses between those respondents who reported fibromyalgia as a pain condition and those that did not and (2) the responses between those respondents who reported osteoarthritis as a pain condition and those that did not. We reported the number and percentage of

**Table 5. Responses to the question: "Which of the following would you like a pain forecast to provide for you?".** Participants could select more than one option.

| Pre-specified response | Number (and percentage) of full population who selected response | Number (and percentage) of subgroup with fibromyalgia who selected response | Number (and percentage) of subgroup with osteoarthritis who selected response |
|---|---|---|---|
| **Information about a pain flare** | 100 (67.6%) | 47 (69.1%) | 34 (69.4%) |
| **Information about fluctuations in pain severity** | 94 (63.5%) | 47 (69.1%) | 27 (55.1%) |
| **Information about pain severity on a scale of 1 to 5** | 70 (47.3%) | 30 (44.1%) | 23 (46.9%) |
| **Information about a period of low/no pain severity** | 51 (34.5%) | 25 (36.8%) | 15 (30.6%) |
| **Other/None** | 13 (8.8%) | 5 (7.4%) | 5 (10.2%) |

respondents in the disease subgroups who selected each response and performed a chi-squared test to test whether these responses were significantly different from those not in the corresponding subgroup. Comparing participants reporting fibromyalgia against those not reporting fibromyalgia gave a *p*-value of 0.2414. For the same question, a chi-squared test of the responses from participants reporting osteoarthritis against those not reporting osteoarthritis gave a *p*-value of 0.2202. Therefore, there is no evidence that the subgroups gave statistically significantly different responses to the population.

Only two relevant free-text responses were provided to this question, both referring to fluctuations in pain severity:

"Will my future pain graphs differ from those in the past?"

"Compare it to half hour ago, a couple of hours ago, yesterday etc. with a higher lower method"

Results of the priority-setting question "*If we could predict a pain flare, what specific information would you want to know*?" are shown in Fig 1. The respondents ranked six statements. Onset of a pain flare was the first priority for 92 (62%) of respondents. Severity of a pain flare was the first or second priority for 74 (50%) of respondents. Pain-related quality of life and variation in other symptoms were given fifth or sixth priority by 104 (70%) and 86 (58%) of respondents, respectively. Responses to this question highlight that the onset of a pain flare and severity of a pain flare are clear priorities for respondents.

In free-text responses, nine respondents highlighted that they also wanted information about the triggers of their pain flare. Specific triggers that were cited were hormonal cycles, weather, environment and mood. One participant wanted information about the acceleration of the pain flare and one wanted information about medication to take during a flare.

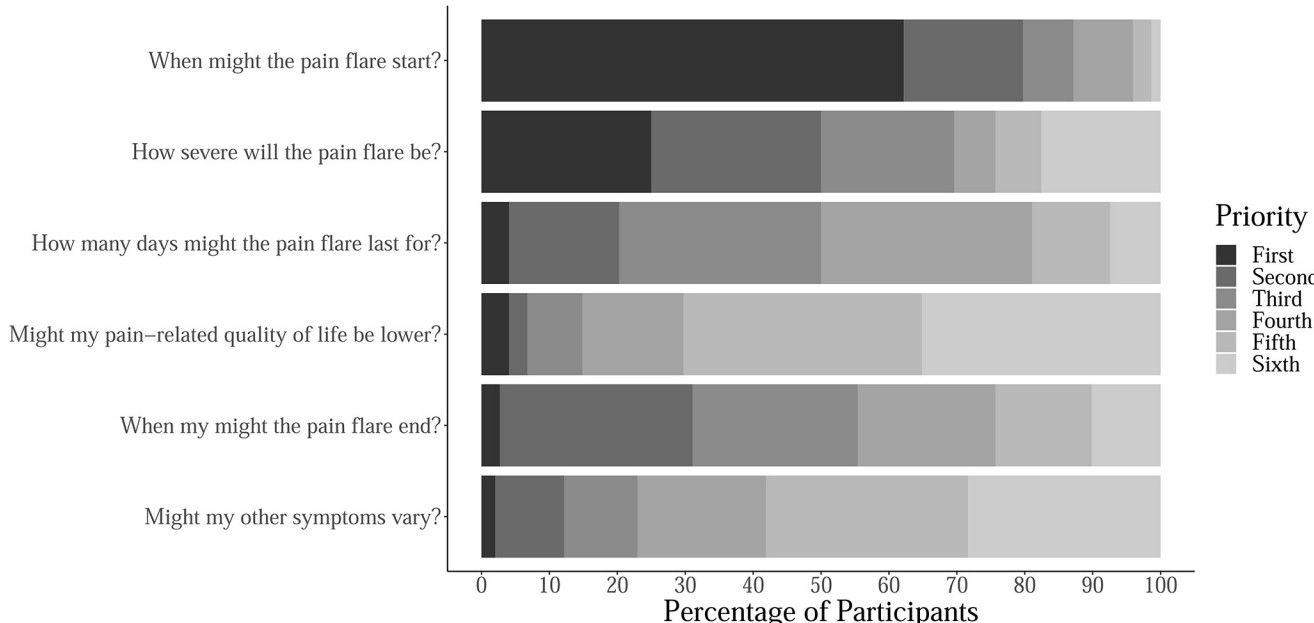

**Fig 1. Survey respondents' priorities relating to pain flares.** Respondents were prompted with the question: "If we could predict a pain flare, what specific information would you want to know?". Percentages of participants ranking each statement as their first, second, third, fourth, fifth or sixth priority are reported.

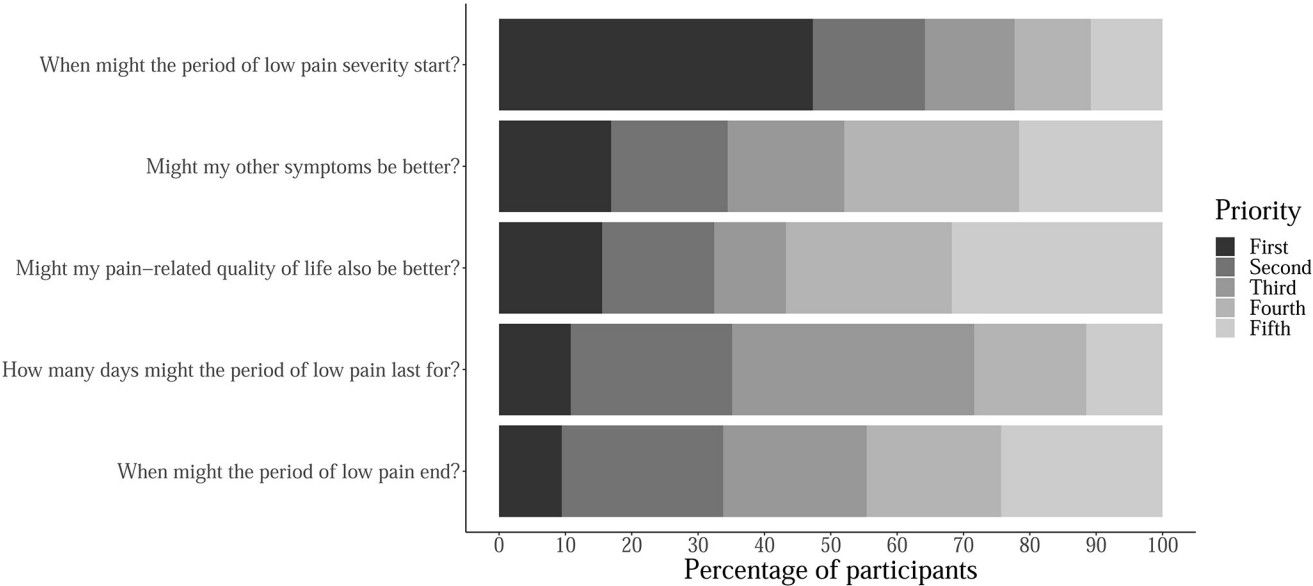

**Fig 2. Survey respondents' priorities relating to periods of low pain severity.** Respondents were prompted with the question: " If we could predict a period of low pain severity, what specific information would you want to know?". Percentages of participants ranking each statement as their first, second, third, fourth or fifth priority are reported.

Results of the priority-setting question "*If we could predict a period of low pain severity, what specific information would you want to know*?" are shown in Fig 2. The respondents ranked five statements. Onset of a period of low pain severity was first priority for 70 (47%) respondents. Respondents did not show great variability among the other responses. First or second priority was given to the duration of the period by 53 (36%) participants, to variation in other symptoms by 52 (35%) participants, to the end of the period by 50 (34%) participants, and to pain-related quality of life by 47 (32%) of participants. There was therefore only a clear priority for information about the onset of the period of low pain severity.

Of the free-text responses, eight respondents referred to wanting information about the triggers of their low pain (e.g. barometric pressure) or variables that they could control (e.g., exercise). One participant wanted to understand how typical their experience is among others, one wanted information about treatments, and one wanted information about specific days.

Participants were asked the multiple-choice question: "*If a pain forecast could provide useful information for you, do you think that you would use a pain forecast?*" (Table 6). Of the 148 respondents, 113 (76%) would use a pain forecast. A chi-squared test of the responses from

**Table 6. Responses to the question: "If a pain forecast could provide useful information for you, do you think that you would use a pain forecast?".**

| Response | Number (and percentage) of full population who selected response | Number (and percentage) of subgroup with fibromyalgia who selected response | Number (and percentage) of subgroup with osteoarthritis who selected response |
|---|---|---|---|
| Definitely not | 0 (0%) | 0 (0%) | 0 (0%) |
| Probably not | 11 (7.4%) | 2 (2.9%) | 4 (8.2%) |
| Might or might not | 24 (16.2%) | 9 (13.2%) | 8 (16.3%) |
| Probably yes | 63 (42.6%) | 27 (39.7%) | 16 (32.7%) |
| Definitely yes | 50 (33.8%) | 30 (44.1%) | 21 (42.9%) |

**Table 7. Responses to the question "What would you use a pain forecast for?" participants could select more than one option.**

| Pre-specified response | Number of participants who selected response | Percentage of participants who selected response |
|---|---|---|
| To help make plans (e.g. shopping, social) | 123 | 83.1% |
| To understand the triggers of my pain | 113 | 76.4% |
| To know when my pain severity might be better/worse | 92 | 62.2% |
| To help plan nonpharmacological interventions | 70 | 47.3% |
| To help choose which medication to take | 46 | 31.1% |
| Other/None | 16 | 10.8% |

participants reporting fibromyalgia against those not reporting fibromyalgia gave a *p*-value of 0.2133. For the same question, a chi-squared test of the responses from participants reporting osteoarthritis against those not reporting osteoarthritis gave a *p*-value of 0.2133. Therefore, there is no evidence that the subgroups gave statistically significantly different responses to the population.

All participants were also asked: "*What would you use a pain forecast for?*" (Table 7). The most common reasons were making plans (123, or 83%) and understanding individual triggers of chronic pain (113, or 76%). In addition, 70 (47%) and 46 (31%) respondents would use a pain forecast to plan pharmacological and nonpharmacological interventions, respectively. Therefore, making plans and understanding triggers are highlighted as the most likely benefits, although planning pharmacological and nonpharmacological interventions may also be of interest to a number of users.

Relevant free-text responses were:

"To improve my overall self management of my conditions"

"To let work know times where I might need time off to recover so that it's not out of the blue for them and they can prepare for me to be off if I need to"

"To analyse the development of my condition"

"Exercise planning"

"To help me understand my condition more"

"To look forward to some good times!"

## Discussion

There are limitations to the PPI activities that should be considered. The representation of different conditions in our survey may have been impacted by the charities that shared the advertisements with their members, perhaps explaining the high prevalence of fibromyalgia among our respondents. This would impact the results if participants with certain conditions had different priorities to other people with chronic pain, and those conditions were over-represented in the surveyed population. However, sensitivity analyses among the subset of participants with fibromyalgia compared to the subset of participants without fibromyalgia, and the subset of participants with osteoarthritis compared to the subset of participants without osteoarthritis found no differences in the reported responses.

Recruitment advertisements clarified that participants would be commenting on a pain forecast, and respondents therefore had an interest in commenting on this topic. A large proportion of our survey participants were interested in a pain forecast and so the results may be

less generalizable to those people who would initially be unsure or less inclined to use a forecast.

Both PPI activities recruited participants online, primarily due to the ongoing COVID-19 pandemic. A pain forecast may be implemented in a future digital intervention and users would then be required to access the internet. However, our work did not include participants who may be less inclined to use the internet for reasons including access and digital literacy. Our findings are therefore not generalizable to these populations and if a digital intervention is developed, work with these populations should be considered.

Previous work has reported on the longer-term prognosis of chronic pain. For example, some studies have followed people with chronic pain over several months or years and identified different trajectories of pain severity among people with chronic pain [45, 46]. Other studies have identified prognostic factors associated with chronic pain outcomes [47]. However, the participants in our focus group and survey have highlighted the importance of forecasting pain on a shorter-term, to support daily activities.

The benefits of a pain forecast extend previous work. Flurey et al. [23] reported that patients expressed frustration at the unpredictability of pain flares and this led to participants cancelling or altering plans. Fullen et al. [26] also found that individuals with chronic low-back pain reported missing out on social events and avoided making commitments, due to the unpredictability of their pain. Our work highlighted that respondents would value a forecast that reduced the unpredictability of their pain, particularly around the timing and severity of pain flares. Our participants highlighted the importance of making plans as a key benefit of a forecast, likely due to the frustration previously reported which has resulted in avoidance of making plans.

The drawbacks highlighted are also consistent with previous work. Among the challenges in collection and analysis of patient generated health data, privacy concerns have previously been highlighted [48], in line with concerns of focus group participants. Any future mobile application of a pain forecast should follow standards of privacy and security [49] and clearly communicate these to users. Furthermore, as pain is widely accepted within the biopsychosocial model [50], and rumination and catastrophizing are associated with increased pain severity [51], concerns around anticipatory anxiety should be thoughtfully considered.

These PPI activities indicated a high level of interest in a pain forecast and our participants were clear that pain features should include the timing of the start of a pain flare, the severity of a pain flare, and fluctuations in pain severity. Future work will develop a statistical pain forecasting model, to predict these identified features. As one of the key benefits of a pain forecast is the identification of triggers, a future model should be interpretable by its users. Drawbacks highlighted in the focus group, such as the impact of anticipatory anxiety should also be considered during the production of a forecast. Wider interest will be determined in the future, based on the uptake of a forecast and continued involvement of stakeholders and evaluation of a forecast will ensure that priorities indicated by participants translate into real value.

## Conclusion

To understand whether individuals with chronic pain would be interested in forecasts of their pain, we conducted two patient and public involvement activities: a focus group of 12 participants and a survey of 148 other participants. These activities were focused on learning about the participants' priorities in the features provided by a hypothetical pain forecast and understanding the perceived benefits that such forecasts would provide. The networks for 14 charities were used to find volunteers for the focus group and the survey.

Participants in the focus group identified the desire to predict the timing of the start of the periods of pain flares (when pain severity increased for a number of days) and their duration. The participants also wanted to know how the pain would affect their quality of life, which would allow them to better make plans, plan pharmacological interventions, and adapt work. Finally, the participants hoped that such hypothetical pain forecasts would help them understand the triggers of their pain, further empowering their ability to manage their disease and its effects. Participants also identified drawbacks to such forecasts, such as the anxiety of knowing of an upcoming pain event, recording their pain-severity scores leading to unnecessary focus on the pain, and the potential sharing of data with employers and the government.

The discussions within the focus group were used to construct a survey asking about the potential benefits and features of a hypothetical pain forecast. A total of 148 individuals responded to the survey. Most of the respondents reported their conditions as fibromyalgia (68, or 46%) and osteoarthritis (49, or 33%). When asked which features they desired in a pain forecast, pain flares (100, or 68%) and fluctuations in pain severity (94, or 64%) were the most commonly reported features. When asked if they would use a pain forecast, most respondents (113, or 76%) said they would. The responses to these questions were not statistically different for the subgroups with fibromyalgia or osteoarthritis relative to the respondents as a whole.

Respondents also identified their priorities for what a pain forecast would include. Most popular features were onset, severity, and duration of a pain flare. When asked what they would use a pain forecast for, respondents replied with their most common reasons: making plans (123, or 83%), understanding individual triggers of chronic pain (113, or 76%), and knowing when their pain severity might be better or worse (92, or 62%). In addition, 70 (47%) and 46 (31%) respondents would use a pain forecast to plan pharmacological and nonpharmacological interventions, respectively.

The results of the focus group and survey indicate the potential for forecasts of pain for those living with chronic pain and what the features of such forecasts might be. Some commercial weather-forecasting companies have smartphone applications that provide pain forecasts based on unspecified algorithms, so this desire is already being met. Nevertheless, our results provide more insight into the benefits that the forecasts would provide to respondents through using these forecasts, as well as their concerns for their mental health knowing about potential future pain flares and their concerns for how the data would be used and who would have access to it.

## Supporting information

**S1 Data. Survey to participants.**
(PDF)

## Acknowledgments

We are grateful for the contributions of the members of the Patient and Public Involvement focus group and the respondents to the survey. We thank Prof. Premalatha Paulsamy and an anonymous reviewer for comments that improved an earlier version of this manuscript.

## Author Contributions

**Formal analysis:** Claire L. Little.

**Investigation:** Claire L. Little.

**Methodology:** Claire L. Little, Katie L. Druce, John McBeth.

**Supervision:** William G. Dixon, David M. Schultz, Thomas House, John McBeth.

**Writing – original draft:** Claire L. Little.

**Writing – review & editing:** Claire L. Little, Katie L. Druce, William G. Dixon, David M. Schultz, Thomas House, John McBeth.

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
