## [Decision Letter · Decision Letter 0]

8 Aug 2023

PONE-D-23-13975What do people living with chronic pain want from a pain forecast? A research prioritization studyPLOS ONE

Dear Dr. Schultz,

Thank you for submitting your manuscript to PLOS ONE. After careful consideration, we feel that it has merit but does not fully meet PLOS ONE’s publication criteria as it currently stands. Therefore, we invite you to submit a revised version of the manuscript that addresses the points raised during the review process.

We look forward to receiving your revised manuscript.

Kind regards,

Shadia Hamoud Alshahrani, PhD

Academic Editor

PLOS ONE

Journal Requirements:

"This work was supported by infrastructure support from the Centre for Epidemiology Versus Arthritis (grant number 21755; https://www.versusarthritis.org/research/our-current-research/our-research-centres/). TH receives funding from the Royal Society (grant number INF/R2/180067; https://royalsociety.org/) and the Alan Turing Institute for Data Science and Artificial Intelligence (https://www.turing.ac.uk/). "

"WGD has received consultancy fees from Google (google.com), and DMS has received consultancy fees from Palta (https://palta.com/), both unrelated to this work. All other authors have declared that no competing interests exist."

Reviewers' comments:

Reviewer's Responses to Questions

**Comments to the Author**

1. Is the manuscript technically sound, and do the data support the conclusions?

Reviewer #1: No

Reviewer #2: Yes

2. Has the statistical analysis been performed appropriately and rigorously? 

Reviewer #1: No

Reviewer #2: Yes

3. Have the authors made all data underlying the findings in their manuscript fully available?

Reviewer #1: No

Reviewer #2: Yes

4. Is the manuscript presented in an intelligible fashion and written in standard English?

Reviewer #1: Yes

Reviewer #2: Yes

5. Review Comments to the Author

Reviewer #1: The article needs major enhancement especially for data presentation, results, intext referencing, major work needs to be done, perhapse rewriting at this point and resubmit it again. Also you need to focus on presenting the data in more than a percentage.

Reviewer #2: Overall, it is a unique study and gives scientific insight into the fraternity on the development of the pain forecast model.

1. Title: The title of the manuscript is appropriate.

2. Introduction: In line no. 36, 37, It would be better to include the most recent references which will give more strength to the need for the study.

3. Study Design: It is unclear from the abstract what is the type and design. It would be helpful to clarify the study design and approach.

4. Methods Section: The methods section lacks important details regarding the study sample selection.

• Is there any matching done according to the sample characteristics

• How the survey question was validated

• In which way, the survey was questionnaire was administered

Please clearly the above quarries.

5. Analysis of data: It is unclear how the data was analyzed. Kindly elaborate on it.

7. Conclusion Section: Consider discussing the implications of the study's findings.

6. PLOS authors have the option to publish the peer review history of their article (what does this mean?). If published, this will include your full peer review and any attached files.

Reviewer #1: No

Reviewer #2: **Yes: **PREMALATHA PAULSAMY

---

## [Author Response · Author response to Decision Letter 0]

11 Sep 2023

Please see the attached document Response2Reviewers-R2.docx.

---

## [Decision Letter · Decision Letter 1]

3 Oct 2023

What do people living with chronic pain want from a pain forecast? A research prioritization study

PONE-D-23-13975R1

Dear Dr. David M. Schultz

We’re pleased to inform you that your manuscript has been judged scientifically suitable for publication and will be formally accepted for publication once it meets all outstanding technical requirements.

Kind regards,

Shadia Hamoud Alshahrani, PhD

Academic Editor

PLOS ONE

**Comments to the Author**

1. If the authors have adequately addressed your comments raised in a previous round of review and you feel that this manuscript is now acceptable for publication, you may indicate that here to bypass the “Comments to the Author” section, enter your conflict of interest statement in the “Confidential to Editor” section, and submit your "Accept" recommendation.

Reviewer #1: All comments have been addressed

2. Is the manuscript technically sound, and do the data support the conclusions?

Reviewer #1: Yes

3. Has the statistical analysis been performed appropriately and rigorously? 

Reviewer #1: Yes

4. Have the authors made all data underlying the findings in their manuscript fully available?

Reviewer #1: Yes

5. Is the manuscript presented in an intelligible fashion and written in standard English?

Reviewer #1: Yes

6. Review Comments to the Author

Reviewer #1: (No Response)

7. PLOS authors have the option to publish the peer review history of their article (what does this mean?). If published, this will include your full peer review and any attached files.

Reviewer #1: No

---

## [Editor Report · Acceptance letter]

5 Oct 2023

PONE-D-23-13975R1 

What do people living with chronic pain want from a pain forecast? A research prioritization study 

Dear Dr. Schultz:

I'm pleased to inform you that your manuscript has been deemed suitable for publication in PLOS ONE. Congratulations! Your manuscript is now with our production department. 

Kind regards, 

on behalf of

Dr. Shadia Hamoud Alshahrani 

Academic Editor

PLOS ONE